Blooming plant species diversity patterns in two adjacent Costa Rican highland ecosystems

Cristóbal-Pérez E. Jacob ejacob@cieco.unam.mx 1 2 3
Barrantes Gilbert 1 3 4
Cascante-Marín Alfredo 1 3 4
Madrigal-Brenes Ruth 1 3 4
Hanson Paul 1 4
Fuchs Eric J. 1 2 3 4
1 Centro de Investigación en Biodiversidad y Ecología Tropical, Universidad de Costa Rica , San José , Costa Rica
2 Laboratorio Nacional de Análisis y Síntesis Ecológica/Escuela Nacional de Estudios Superiores Unidad Morelia, Universidad Nacional Autónoma de México , Morelia , Michoacán , Mexico
3 Laboratorio Binacional de Análisis y Síntesis Ecológica UNAM-UCR, Universidad Nacional Autónoma de México, Universidad de Costa Rica , Morelia , Michoacán , Mexico
4 Escuela de Biologia, Universidad de Costa Rica , San Jose , Costa Rica
Singh Randeep
Electronic publication date: 2023 Jan 12
Publication date: 2023
Volume: 11
Electronic Location ID: e14445
Received 2022 Sep 3; Accepted 2022 Nov 1
Copyright: ©2023 Cristóbal-Pérez et al.
Copyright year: 2023
Copyright holder: Cristóbal-Pérez et al.
License: This is an open access article distributed under the terms of the Creative Commons Attribution License, which permits unrestricted use, distribution, reproduction and adaptation in any medium and for any purpose provided that it is properly attributed. For attribution, the original author(s), title, publication source (PeerJ) and either DOI or URL of the article must be cited.
License URL: https://creativecommons.org/licenses/by/4.0/

Keywords: Beta diversity, Endemism, Floral syndromes, Paramo, Plant species composition, Montane forest

Funding: UCR Vicerrectoría de Investigación-UCR C1460 C0-517 C0-068 B6-A32 MICIT-CONICYT FI-040B-19 E. Jacob Cristóbal-Pérez was supported by a postdoctoral fellowship from UCR. Financial support for this project was provided by Vicerrectoría de Investigación-UCR (C1460, C0-517, C0-068, B6-A32) and MICIT-CONICYT (FI-040B-19). There was no additional external funding received for this study. The funders had no role in study design, data collection and analysis, decision to publish, or preparation of the manuscript.

==============================
The Costa Rican Paramo is a unique ecosystem with high levels of endemism that is geographically isolated from the Andean Paramos. Paramo ecosystems occur above Montane Forests, below the permanent snow level, and their vegetation differs notably from that of adjacent Montane Forests. We compared the composition and beta diversity of blooming plant species using phenological data from functional plant groups (i.e., insect-visited, bird-visited and insect + bird-visited plants) between a Paramo and a Montane Forest site in Costa Rica and analyzed seasonal changes in blooming plant diversity between the rainy and dry seasons. Species richness was higher in the Montane Forest for all plant categories, except for insect-visited plants, which was higher in the Paramo. Beta diversity and blooming plant composition differed between both ecosystems and seasons. Differences in species richness and beta diversity between Paramo and the adjacent Montane Forest are likely the result of dispersal events that occurred during the last glacial period and subsequent isolation, as climate turned to tropical conditions after the Pleistocene, and to stressful abiotic conditions in the Paramo ecosystem that limit species establishment. Differences in blooming plant composition between both ecosystems and seasons are likely attributed to differential effects of climatic cues triggering the flowering events in each ecosystem, but phylogenetic conservatism cannot be discarded. Analyses of species composition and richness based on flowering phenology data are useful to evaluate potential floral resources for floral visitors (insects and birds) and how these resources change spatially and temporarily in endangered ecosystems such as the Paramo.

Introduction

A notable characteristic of tropical highland landscapes is the presence of well defined ecotones between adjacent ecosystems at high elevations (Vuilleumier & Simberloff, 1980; Sarmiento, 2021). This sudden change in the vegetation physiognomy is attributed mainly to differences in climatic and edaphic conditions (Luteyn, 2005). The highest mountain environments above the treeline are unsuitable habitats for most organisms that inhabit adjacent tropical forests at lower elevations (Luteyn, 2005; Körner, 2021). It has been suggested that changes in the composition of plant communities along altitudinal gradients may be determined by environmental filtering, since increasing altitudes are often associated with harsh conditions for life (Laiolo & Obeso, 2017). Hence, only a relatively low number of species have been capable of adapting to the prevailing abiotic conditions at high altitudes, resulting in a general decline in species richness but an increase in endemism (Billings, 1974; Rada, Azócar & García-Núñez, 2019; Madriñán, Cortés & Richardson, 2013). In the Neotropics, the Paramo exemplifies a high elevation ecosystem; this habitat is typically composed of low herbaceous and shrubby vegetation whose physiognomy drastically contrasts with the arboreal vegetation that dominates the adjacent Montane Forests (Smith & Young, 1987; Luteyn, 2005).

Most of the neotropical Paramos (including the Puna) are found in South America and cover a large proportion of the highlands of the Andes mountain range (Madriñán, Cortés & Richardson, 2013). In Central America, the Paramo vegetation is restricted to highly isolated and small natural fragments on the highlands of the Talamanca mountain range that extends from Costa Rica to western Panama (Kappelle & Horn, 2016). As a result, South American Paramos have been the focus of research on a variety of topics, including plant physiology (Rada, Azócar & García-Núñez, 2019), avian evolution (Vuilleumier, 1969), vegetation (Valencia et al., 2018) and butterfly distribution (Pyrcz et al., 2016); whereas research in Central American Paramo ecosystem is still limited (Körner, 2021). A book published by Kappelle & Horn (2005) included information on the natural history of many taxa from the Costa Rican Paramo, but information on the ecology and evolution of most taxonomic groups was anecdotal or based on non-systematic samplings.

The species diversity turnover of plants and other taxonomic groups along altitudinal gradients has been studied worldwide and, in general, richness in all groups decreases with elevation, but endemism increases (Wolda, 1987; Navarro, 1992; Lieberman et al., 1996; Vetaas & Grytnes, 2002; Khuroo et al., 2011; Steinbauer et al., 2016; Monro, Bystriakova & Gonzalez, 2017). There are also changes in abiotic conditions such as a reduction in availability of surface area, atmospheric pressure, air temperature, and increasing UV radiation at higher elevations (Körner, 2007). For sessile organisms such as plants, these environmental gradients impose severe constraints on growth, survival, flowering and fruiting phenology, which may influence the feeding behavior and reproduction of associated organisms such as insects and birds. Tropical highland ecosystems are also characterized by a marked seasonal variation in rainfall and daily temperatures between the dry and rainy seasons (Sarmiento, 1986). Seasonality is a proximal factor that can regulate plant phenology (Borchert, 1983; Reich & Borchert, 1982; Reich & Borchert, 1984; Cavelier et al., 1992; Smith & Young, 1987), and therefore may constrain floral resource availability for floral visitors.

In the Costa Rican highland ecosystems, plant richness also declines rapidly with elevation, particularly at mountain summits (Lieberman et al., 1996; Estrada & Zamora, 2004; Barrantes, Chacón & Hanson, 2019; Monro, Bystriakova & Gonzalez, 2017). However, information on the dynamics of floral resources availability (i.e., flowering phenology patterns) at the community level remains undocumented. Patterns of plant reproductive phenology may be related to the variation in floral resource availability and changes in the community composition of floral visitors throughout the year.

Flowering plants may be classified into different pollination syndromes based on a set of floral traits (e.g., morphology, color, odor, size, rewards, and anthesis time) (Faegri & Van der Pijl, 1979; Rosas-Guerrero et al., 2014). Most plant species inhabiting highland tropical ecosystems can be classified into insect-pollinated (bees and flies), bird-pollinated and insect + bird-pollinated pollination syndromes. Evidence suggests that as elevation increases, flower-visitor diversity, population abundance, and foraging activity decreases (Arroyo, Armesto & Villagran, 1981; Gómez-Murillo & Cuartas-Hernández, 2016). However, there is no information on the availability of floral resources in relation to the type of floral visitors in Central American highland ecosystems.

This study has a twofold objective: to determine differences in floral resources availability in terms of blooming plant composition and diversity between the two high-elevation ecosystems in Costa Rica (Paramo and Montane Forest), and to describe their variation in resource availability for insects and birds between the dry and rainy seasons. We predict significant differences in community composition between the Paramo and the adjacent Montane Forest, with higher species richness and beta-diversity of blooming plants in Montane Forests, due to the large number of endemic species present in the Paramo and the reduction in species richness as elevation increases. We also predict a higher diversity of the blooming plant community in the rainy season, in both ecosystems, due to milder temperatures and higher water availability compared to more severe conditions prevalent during the dry season.

Materials and Methods

Study area

We selected two study sites in the highlands of the Costa Rican Talamanca mountain range: the Cerro de la Muerte Biological Station (CMBS) and the Quetzales National Park (QNP) (Fig. 1). The CMBS is a Montane Forest at an elevation of 3100 m asl (09°33′N; 83°44′W) and the QNP is a Paramo habitat at 3400 m asl (Fig. 1). The two sites are separated by 2 km. The region’s average annual precipitation is 2500 mm, with a relatively dry period from mid-November to April, and a mean annual temperature of 11 °C for the CMBS and 7.6 °C in the QNP (Herrera, 2005). During the day, temperatures fluctuate dramatically, particularly in the Paramo (−5 °C to 35 °C) (Herrera, 2005). Montane Forests are dominated by oaks with abundant epiphytes and shrubs (e.g., Ericaceae, Asteraceae, Onagraceae) (Calderón-Sanou et al., 2019). Meanwhile, the Paramo is dominated by a herbaceous stratum, with a large diversity of Asteraceae and Poaceae, and scattered patches of shrubs with species mainly in the Ericaceae, Asteraceae, and Hypericaceae (Vargas & Sánchez, 2005).

Figure 1 Map of the region of the Costa Rican Talamanca mountain region, showing the study sites and the elevation above sea level (asl): the Cerro de la Muerte Biological Station (CMBS) and the Quetzales National Park (QNP).

Sampling

In each study site, we established a 2 km by 10 m transect and counted the number of individual blooming plants per species per month, during a 30 month period (February 2019 to August 2021). We classified each plant species into insect-pollinated (bee-pollinated and fly-pollinated), bird-pollinated (hummingbirds) and insect + bird-pollinated types, based on their morphology and floral reward following Barrantes (2005) and Rosas-Guerrero et al. (2014). We did not include wind-pollinated species, such as oaks (Fagaceae), grasses and sedges (Poaceae and Cyperaceae, respectively). We defined the flowering peak for the whole community at each site and for each plant category (insect-pollinated plants, insect + bird-pollinated plants, bird-pollinated plants) as the month(s) fitting into the third quartile; if a sequence of months all met this requirement, we chose the month with the highest number of flowering individuals.

Statistical analyses

We compared species richness between the Montane Forest and the Paramo by means of rarefaction curves with 95% confidence intervals, using the function specaccum in the R package vegan (Oksanen et al., 2020). This method controls for differences in sample size by estimating the expected species richness of a random subsample of individuals (Gotelli & Graves, 1996).

To compare the plant community composition between sites, we used a non-metric multidimensional scaling (NMSD) based on a Bray-Curtis dissimilarity matrix with 1000 permutations. We then conducted a distance-based Permutational Multivariate Analyses of Variance (PERMANOVA) as implemented in the adonis function in the R package vegan (Oksanen et al., 2020). For this analysis, we included site (Montane Forest and Paramo), season (Dry and rainy seasons), and their interaction as independent factors and the distance matrix as the response variable.

Subsequently, we compared beta diversity between the two sites, measured as the mean dissimilarity non-Euclidean distance of each individual observation to the mean of all observations (centroid) calculated in multidimensional space, as implemented by the betadisper function (Anderson, Ellingsen & McArdle, 2006; Oksanen et al., 2020). This function is used to test the homogeneity of variances between sites or treatments. However, PERMANOVA is unaffected by the heterogeneity of variances for balanced designs (Anderson & Walsh, 2013), as is the case in this study (equal sampling at both sites). Therefore, we used the betadisper function to test for differences in beta-diversity between sites, as has been used in other studies (Oksanen et al., 2020). We used the vegan package (Oksanen et al., 2020) in the R statistical language for all analyses (R Development Core Team, 2021).

Results

We recorded the flowering phenology of 91 species in 41 families: 72 species in the Montane Forest and 65 in the Paramo; 46 of these species were present at both sites. Based on our rarefaction analysis, the species richness of blooming plants was higher in the Montane Forest (Fig. 2). Similarly, the richness of plants pollinated by insects + birds and by birds only was higher in the Montane Forest; however, richness of insect-visited plants was higher in the Paramo site (Figs. S1a–S1c). This indicates that both ecosystems offer a great diversity of food resources for different pollinator guilds. More resources were available for hummingbirds in the Montane Forest, while insects seem to benefit more from plants in the Paramo ecosystem.

Figure 2 Sample-based rarefaction curves with 95% confidence intervals for flowering plant richness in the Montane Forest (green lines) and the Paramo (golden lines) ecosystems from the Costa Rican Talamanca mountain range.

Data are from flowering censuses from February 2019 to August 2021.

The number of blooming plant species varied over time (Fig. 3). All blooming plant species in both ecosystems peaked during the dry season (Fig. 3), but insect-pollinated plants had flowering peaks at the beginning (May) and the second half of the rainy season (September-October) (Fig. S2a). Insect + bird and bird-pollinated plant categories did not show a clear seasonal pattern (Figs. S2b–S2c); on the contrary, floral resources in these two plant categories varied little throughout the year. In the case of bird-pollinated plant species, the number of blooming species was always higher in the Montane Forest than in Paramo (Fig. S2d).

Figure 3 Number of blooming plant species in the Montane Forest (green dots) and the Paramo (golden dots) recorded during the study period of February 2019 to August 2021 in the Costa Rican Talamanca mountain range.

The solid black lines above the x-axis indicate the dry season months.

The multidimensional scaling distances showed that species composition differed between sites (Montane Forest and Paramo), seasons (dry and rainy), and their interaction for all plant categories (i.e., all blooming plant species, insect-visited plants, insect + bird-visited plants and bird-visited-plants) (Table 1; Fig. S3; Table S1). In all cases, the site explained the largest fraction of the variance, followed by season, and then their interaction (Table 1), though there is still a large portion of the variance that is not explained by the factors included in the model. This is expected since phenological cues are multifactorial, and their synergistic effect is not yet fully understood (Satake, Nagahama & Sasaki, 2022). The changes in species composition between the rainy and dry seasons are more pronounced in the Montane Forest than in the Paramo, for all blooming plant species (Fig. 4). However, this pattern is reversed for bird-pollinated plant species, where species composition differences between the dry and the rainy season are greater in the Paramo compared to the Montane Forest (Fig. S4).

Table 1 Non-parametric PERMANOVA based on Bray–Curtis distances for all blooming plants at two sites (Montane Forest and the Paramo), two seasons (dry and rainy), and their interaction.

All blooming plants (MSD/Bray − Stress = 0.98)	
Factor	df	SS	R2	F	P	
Site	1	5.03	0.38	48.55	0.001	
Season	1	1.46	0.11	14.11	0.001	
Site*season	1	0.87	0.06	8.37	0.001	
Residual	58	5.99	0.45			
Total	61	13.36	1.00			

Figure 4 Effect of site and season (RainyM, Montane Forest—rainy season; DryM, Montane Forest—dry season; RainyP, Paramo—rainy season; DryP, Paramo—dry season) on the beta diversity of blooming plant species, in the Costa Rican Talamanca mountain range.

The analysis was performed using the betadisper function in R. Each dot represents the mean non-Euclidean distance of blooming plants at a particular sampling date relative to the centroid of all samplings on the two first PCA components.

Beta diversity for each plant pollination type differed between the Montane Forest and the Paramo for all categories of blooming plants (Table 2, Fig. 4; Figs. S4a–S4d). This suggests that particular factors have shaped each ecosystem, such as climatic conditions and underlying historical factors (e.g., colonization-dispersal events) and influenced the beta diversity of blooming plants differently.

Table 2 Comparison of beta diversity for blooming plants between Montane Forest and Paramo forest in Costa Rica, based on the “betadisper” function (Oksanen et al., 2020).

Factor	df	SS	MS	F	P	
All blooming plants	
Site	1	0.03	0.03	9.48	0.003	
Residual	60	0.21	0.00			
Insect + bird-visited plants	
Site	1	0.03	0.03	9.26	0.002	
Residual	60	0.22	0.00			
Insect-visited plants	
Site	1	0.05	0.05	11.96	0.002	
Residual	60	0.27	0.00			
Bird-visited plants	
Site	1	0.05	0.05	3.83	0.057	
Residual	60	0.76	0.01			

Discussion

Our results show differences in species composition and diversity of plants between two adjacent ecosystems at tropical high elevations. Local and regional environmental traits, and historical events likely act synergistically to produce the differences observed (Simpson, 1975; Hooghiemstra et al., 1992; Islebe, Hooghiemstra & Van der Borg, 1995; Islebe, Hooghiemstra & Van’t Veer, 1996; Sklenář, Dušková & Balslev, 2011; Barrantes, 2009). In comparison to the adjacent Montane Forest, the Paramo has a lower richness of flowering species. The study sites are geographically adjacent and separated by 2 km; however, the relatively small change in elevation (∼400 m) becomes a determinant factor in shaping species composition differences. Consequently, temporal turnover (beta diversity) of blooming plants also differed between ecosystems and such differences are likely related to the uniqueness of the Costa Rican Paramo vegetation (Cleef & Chaverri, 1992). The evolution of a unique vegetation in the Costa Rican Paramo, which differs notably from the adjacent Montane Forest, could be the result of several factors: (a) the plant dispersal events that occurred during the late Pleistocene (Simpson & Neff, 1992; Sklenář, Dušková & Balslev, 2011; Londoño, Cleef & Madriñán, 2014), (b) the geographic isolation when climate changed after the Pleistocene, and (c) the prevalence of cold climatic conditions at the summit of the Talamanca Mountain range. Vicariance driven by the climate shifts after the Pleistocene in conjunction with topographic isolation, has shaped the evolution of several plant clades within the Andean cordilleras (Simpson, 1975; Luebert & Weigend, 2014). For instance, a possible explanation for the rapid radiation of the common Valeriana and Hypericum species in the Andean Paramo, as well as the species present in the Costa Rican Paramo, is the repeated fragmentation-isolation process, as a consequence of the Pleistocene climatic fluctuations in a topographically complex region (Moore & Donoghue, 2007; Nürk, Scheriau & Madriñán, 2013).

Temporal variation in floral resources imposes a constraint on plant–pollinator interactions (Hegland & Boeke, 2006; Fuchs, Ross-Ibarra & Barrantes, 2010; Encinas-Viso, Revilla & Etienne, 2012; Bagella et al., 2013). Our results showed that the flowering phenology of all groups of plants differed between dry and rainy seasons. When we analyzed the entire blooming plant community as a whole (i.e., insect-visited, bird-visited, and insect + bird-visited plants) flowering peaks occurred in the dry season, in contrast to insect-visited plants, whose flowering peak occurred in the rainy season. Such differences are often associated with the response of different groups of plants to different environmental cues (Arroyo, Armesto & Villagran, 1981; Defila & Clot, 2005; Davies et al., 2013; Chmura et al., 2018; Satake, Nagahama & Sasaki, 2022). In other seasonal ecosystems, it has been suggested that water acquisition and storage strategies associated with growth form are related to different temporal patterns of flowering (Cortés-Flores et al., 2017). For example, flowering of herbaceous species occurs during the rainy season, while flowering in trees and shrubs can occur during both rainy and dry seasons (Frankie, Baker & Opler, 1974; Batalha & Martins, 2004; Cortés-Flores et al., 2017). The flowering phenology patterns that we observed are consistent with the assumption that in seasonal tropical ecosystems, insect pollinators are more abundant during the rainy season, when more floral resources are available (Southwood, Brown & Reader, 1979; Siemann et al., 1998; Ramírez, 2006; Souza et al., 2018). At least one hummingbird species is active year-round in our study site, a pattern recognized in other tropical studies, which reported continuous hummingbird activity across the year (Barrantes, 2005; Abrahamczyk et al., 2011). The presence of a particular floral visitor functional group throughout the year can be explained by the staggered flowering phenologies of plant species in tropical communities, as shown in this study (Lopezaraiza-Mikel et al., 2013; Lobo et al., 2003; Abrahamczyk et al., 2011; Meléndez-Ramírez, Ayala & González, 2016).

An important difference between the Costa Rican Paramo and Andean Paramos is that in Costa Rica, this ecosystem covers only a small and isolated area at the summit of the Talamanca mountain range. Such conditions make this site unique and susceptible to threats imposed by climate change and human intervention. Projections on climate change indicate that temperatures and the length of dry season will increase in the highlands, seriously threatening this ecosystem in Central and South America (Karmalkar, Bradley & Diaz, 2008; Lyra et al., 2017; Freeman et al., 2018). In Costa Rica, the Paramo ecosystem is protected within national parks, but despite this level of protection, they are subject to a wide range of pressures from human activities, such as anthropogenic fires, the construction of communication towers, and agricultural and urban expansion around protected areas, as well as the invasion of exotic plant species (Chaverri & Esquivel-Garrote, 2005) which, in addition to climate change, seriously threaten this unique ecosystem.

Conclusions

We conclude that richness and beta diversity of blooming plant species differed between the Paramo and the adjacent Montane Forest, and such differences are likely a consequence of historical events (e.g., dispersal promoted by changes in climate), and the edaphic and climatic conditions prevailing in the study region. Floral resource availability differed between the two seasons (dry and rainy), due to differences in climatic conditions (Körner, 2021) that may act as environmental cues that trigger the phenological patterns in different plant species; however, a phylogenetic effect (e.g., related plant species flowering at the same time due to common ancestry) cannot be discarded (Davies et al., 2013). Our findings also showed that the composition and diversity of floral resources for insects and birds are lower in the Paramo than in the Montane Forest. This supports the idea that resource depletion may limit the use of the Paramo for nectar-feeding birds and insects (Janzen et al., 1976; Barrantes, 2005; Fuchs, Ross-Ibarra & Barrantes, 2010). This study showed that analyses of species composition and richness based on flowering phenology data are useful in evaluating potential floral resources for floral visitors (insects and birds), and how these resources change spatially and temporarily in these endangered ecosystems.

Supplemental Information

Supplemental Information 1 Supplemental figures and table

Click here for additional data file.

Supplemental Information 2 Raw data of phenology of 30 months of blooming plant species from Costa Rican montane forest and paramo

Click here for additional data file.

We are indebted to the late Federico Valverde for allowing us to conduct this research at the Cerro de la Muerte Biological Station. We thank Zulema Carrillo-Aldape for assistance in preparing the map.

Additional Information and Declarations

Competing Interests

Author Contributions

Field Study Permissions

Data Availability

The authors declare there are no competing interests.

E. Jacob Cristóbal-Pérez conceived and designed the experiments, performed the experiments, analyzed the data, prepared figures and/or tables, authored or reviewed drafts of the article, and approved the final draft.

Gilbert Barrantes conceived and designed the experiments, performed the experiments, analyzed the data, prepared figures and/or tables, authored or reviewed drafts of the article, and approved the final draft.

Alfredo Cascante-Marín conceived and designed the experiments, performed the experiments, authored or reviewed drafts of the article, and approved the final draft.

Ruth Madrigal-Brenes conceived and designed the experiments, authored or reviewed drafts of the article, and approved the final draft.

Paul Hanson conceived and designed the experiments, authored or reviewed drafts of the article, and approved the final draft.

Eric J. Fuchs conceived and designed the experiments, performed the experiments, authored or reviewed drafts of the article, and approved the final draft.

The following information was supplied relating to field study approvals (i.e., approving body and any reference numbers):

The Los Quetzales National Park (MINAE) for granted permission for this investigation (SINAC-ACC-D-RES-1410-2020, SINAC-ACLA-P-D-365-2020, SINAC-ACC-D-re-835-2021).

The following information was supplied regarding data availability:

Raw data of phenology of 30 months of blooming plant species from Costa Rican montane forest and paramo are available in the Supplemental Files.

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
