# Peer review of "Blooming plant species diversity patterns in two adjacent Costa Rican highland ecosystems"

_PeerJ, doi:10.7717/peerj.14445_

## Round 0.1 · original submission · Minor Revisions

Both reviewers suggested some changes. They are very important.

·

Basic reporting

Dear author, I have proceeded to review the manuscript entitled “Blooming plant species diversity patterns in two adjacent Costa Rican highland ecosystems" whose aims was to compared the composition and beta diversity of blooming plant species using phenological data from functional plant groups (i.e., insect-visited, bird-visited and insect+bird-visited plants) between a Paramo and a Montane Forest site in Costa Rica and analyzed seasonal changes in blooming plant diversity between the rainy and dry seasons. Prior to further processing if the editor deems so; it is necessary to cover the following comments very thoroughly:

- The abstract is well structured.
- The introduction addresses important information that puts in context the key words and variables used in the study.
- In the final part of the introduction it would be better if the authors did not narrate in the first person the objective of the study and expected results.

Experimental design

Regarding the experimental design:

- In the study area section it is necessary to include a location figure and the sampling sites.
- I would recommend marking the difference between discussion and conclusions.
- As much as possible the discussion should be more synthesized, try not to be redundant in the discussion.
- Be precise in the conclusions.
- For a correct discussion and conclusion consider the objectives established in your study and try to generate a single paragraph for each of them.

Validity of the findings

no comment

Additional comments

no comment

Reviewer 2 ·

Basic reporting

The manuscript is well written, it presents a concrete objective and clear predictions, a good description of the statistical methods and analyses, as well as concrete results and a good discussion.

Experimental design

Just have a suggestion as to the description of the methods, and where I think it's not very clear, and it relates to how the plant phenology observations were carried out. Authors mention that transects were made, and that in these flowering plants (individuals) were recorded, and that they were assigned categories according to the functional groups of pollinators. But, I consider that it is not very clear if it is only the presence-absence data, or if they had a more quantitative data to evaluate the phenology of the plants.

Validity of the findings

Regarding the statistical analyses, I consider that they are the correct ones and with those presented in the supplementary material they contribute in a splendid way to complement them.
The results are very relevant since they show not only the phenological rhythms of plants in these two sites, very important given their low geographical representation, and their enormous biodiversity, but also that they are among the most threatened ecosystems and most exposed to climatic change. The information provided by this research is extremely relevant to understand these environments a little more and how susceptible they can be, not only to the loss per se of biodiversity, but also to functional aspects, in this case of the pollination.

Additional comments

No comments

---

## Round 0.2 · accepted · Accept

This is an interesting study, and the authors have made the revisions as per suggestions, so my recommendation is to accept.

Reviewer 2 ·

Basic reporting

No comment

Experimental design

Authors include the suggestions

Validity of the findings

No comment

Additional comments

No comment